# Phenotypic and Functional Alterations of Immune Effectors in Periodontitis; A Multifactorial and Complex Oral Disease

**DOI:** 10.3390/jcm10040875

**Published:** 2021-02-20

**Authors:** Kawaljit Kaur, Shahram Vaziri, Marcela Romero-Reyes, Avina Paranjpe, Anahid Jewett

**Affiliations:** 1Division of Oral Biology and Oral Medicine, School of Dentistry and Medicine, Los Angeles, CA 90095, USA; drkawalmann@g.ucla.edu (K.K.); ajewett@dentistry.ucla.edu (S.V.); 2Department of Neural and Pain Sciences, University of Maryland, Baltimore, MD 21201, USA; mromero@umaryland.edu; 3Department of Endodontics, University of Washington, Seattle, DC 98195, USA; avina@u.washington.edu; 4The Jonsson Comprehensive Cancer Center, UCLA School of Dentistry and Medicine, Los Angeles, CA 90095, USA

**Keywords:** cell death, periodontitis, CD69, oral blood, *F. nucleatum*, NFκB, IFN-γ

## Abstract

Survival and function of immune subsets in the oral blood, peripheral blood and gingival tissues of patients with periodontal disease and healthy controls were assessed. NK and CD8 + T cells within the oral blood mononuclear cells (OBMCs) expressed significantly higher levels of CD69 in patients with periodontal disease compared to those from healthy controls. Similarly, TNF-α release was higher from oral blood of patients with periodontal disease when compared to healthy controls. Increased activation induced cell death of peripheral blood mononuclear cells (PBMCs) but not OBMCs from patients with periodontal disease was observed when compared to those from healthy individuals. Unlike those from healthy individuals, OBMC-derived supernatants from periodontitis patients exhibited decreased ability to induce secretion of IFN-γ by allogeneic healthy PBMCs treated with IL-2, while they triggered significant levels of TNF-α, IL-1β and IL-6 by untreated PBMCs. Interaction of PBMCs, or NK cells with intact or NFκB knock down oral epithelial cells in the presence of a periodontal pathogen, *F. nucleatum,* significantly induced a number of pro-inflammatory cytokines including IFN-γ. These studies indicated that the relative numbers of immune subsets obtained from peripheral blood may not represent the composition of the immune cells in the oral environment, and that orally-derived immune effectors may differ in survival and function from those of peripheral blood.

## 1. Introduction

Periodontitis is an inflammatory disease affecting the supporting tissues of the tooth, and is characterized by a wide range of clinical, microbiological and immunological manifestations [1]. The hallmark of periodontitis is the gradual destruction of supporting tissues, which are composed of gingival and periodontal connective tissue, cementum and alveolar bone [2,3]. Several established causes of periodontitis relate to the imbalance in microbial organisms, heightened host’s inflammatory and immune responses and a series of environmental and genetic factors [1,4,5]. Limited knowledge is available about the function, biology and phenotypic properties of oral blood and immune cells infiltrating the gingival tissues, and their comparison with immune cells within the peripheral blood; however, it is clear that pathogenesis of periodontitis is complex and involves both innate and adaptive immune responses [6,7]. In gingiva local inflammatory responses induced by the interaction of stromal cells with the immune effectors in the presence of oral bacteria activate innate immune responses resulting in the release of an array of cytokines and chemokines responsible for continuous recruitment of inflammatory cells to the gingival tissues, and the establishment of chronic inflammation [8]. In addition, many NF-kB-induced pathways are also known to be involved in periodontal diseases [9,10].

It has been reported that both T and B cells are present in periodontal tissues, and both gingival tissue-derived T and B cells were shown to be at a more advanced stage of the cell cycle than peripheral blood T and B cells, indicative of activation within the tissues [2]. Much less is known regarding the NK cells in periodontal diseases, which are known to be the regulators of adaptive immunity [11,12]. The adaptive immune responses in particular, CD4+ T cells and the proinflammatory cytokines IFN-γ and TNF-α are important effectors of bone loss in periodontal disease [13,14,15]. TNF-α was found to be higher in periodontal tissues in comparison to those from healthy individuals [16]. IFN-γ is primarily produced by activated T and NK cells and plays an important role in host defense. IFN-γ knock-out mice were shown to have a decreased bone loss in periodontal disease; however, due to the significance of this cytokine in bacterial defense, the function of this cytokine is very complex and is not clearly known in periodontal disease [17]. T cells from periodontitis patients also express higher secretion of IFN-γ compared to T cells from healthy individuals [18].

Oral microorganisms can induce activation of NK and T cells resulting in the secretion of pro-inflammatory cytokines IL-1β and TNF-α by these cells [19]. IL-1β and TNF-α can also induce the endothelial cells in the blood vessels to express higher ICAM-1 (CD54) and other adhesion molecules [20], allowing more leukocytes to migrate into the periodontal tissues. Previously, in the established lesions of periodontitis, massive accumulations of leukocytes primarily T and B cells were observed [2]. Despite the presence of immune cells, the disease fails to resolve if bacteria remain in the gingival sulcus. The potent activation and induction of cell death by bacteria in epithelial cells may also recruit more immune effectors to the sites of infection, thereby activating immune cells to prevent access of pathogenic oral microorganisms to deeper tissues. Indeed, we and other laboratories reported that F. *nucleatum* is capable of inducing cell death of immune effectors as well as oral keratinocytes in in vitro culture conditions [21]. Persistent recruitment and activation of immune effectors due to continuous activation and death of oral epithelial cells by the oral organisms may result in the increased survival of immune effectors and further the contribution of activated lymphocytes to increased tissue damage and inflammation.

In this paper we investigated the cell surface receptor expression, activation markers, cytokine secretion and cell death profiles of mononuclear cells obtained from peripheral blood, oral blood and gingival tissues of healthy individuals and patients with periodontitis when they were left untreated or treated with interleukin 2 (IL-2), interferon-gamma (IFN-γ) and phorbol myristate acetate (PMA)/ionomycin (I). Since genetic factors, primarily contributed by mutations seen in the pro-inflammatory cytokines such as IL-1β, TNF-α and many others, have been identified to be associated with periodontal disease, we studied NFkB signaling pathway in keratinocytes involved in the regulation of many pro-inflammatory cytokines in order to understand the complex interaction between the immune cells, keratinocytes and oral bacteria.

## 2. Materials and Methods

### 2.1. Cell Lines, Reagents and Antibodies

Mononuclear cells isolated from healthy individuals’ and periodontitis patients’ peripheral and oral blood were cultured in RPMI 1640 supplemented with 1% sodium pyruvate, 1% non-essential amino acids, 1% glutamine, 1% penicillin-streptomycin (Life Technologies, Carlsbad, CA, USA) and 10% fetal bovine serum (FBS) (Gemini Bio-Product, West Sacramento, CA, USA). HEp2 tumor cell lines were obtained from ATCC and maintained on DMEM media (Life Technologies, CA, USA) supplemented with 10% FBS. Oral squamous carcinoma cells (OSCCs) were maintained in RPMI 1640 supplemented with 10% FBS. Human oral keratinocytes (HOK-16B) were cultured in keratinocyte growth medium (KGM) supplemented with 4% bovine pituitary extract, 1% hydrocortisone, 1% gentamycin-sulfate, 1% bovine insulin and 1% epidermal growth factor obtained from Cambrex-Bio (Walkersville, MD, USA). Propidium iodide (PI), phorbol 12-myristate 13-acetate (PMA) and ionomycin were purchased from Sigma (St Louis, MO, USA). *Fusobacterium nucleatum* (PK1594) was obtained from Paul Kolenbrander, National Institutes of Health. Recombinant human IL-2 and IFN-γ were obtained from NIH-BRB. IFN-γ was obtained from Peprotech (Piscataway, NJ, USA). Anti-CD16 mAb, as well as all of the human ELISA kits and flow cytometric antibodies were purchased from Biolegend (CA, USA). Multiplex assay kits were purchased from Millipore (Billerica, MA, USA). pRcCMV-IκB(S32AS36A) and pRcCMV vector alone were generated in our laboratory.

### 2.2. Donor Selection and Diagnostic Criteria

Oral blood and gingival tissues were obtained from consenting donors who were undergoing periodontal surgery at the UCLA school of dentistry, Los Angeles, CA, USA. Patients were classified as having periodontal disease on the basis of bleeding index, attachment loss, probing depth (6 sites/tooth) and radiographic examinations. Those classified as having periodontal disease had each of the following; probing depth of greater than 5 mm, spontaneous bleeding on probing, clinical attachment loss and radiographic evidence of severe alveolar bone loss. Donors were diagnosed as healthy individuals if they demonstrated a probing depth of equal or less than 4 mm, no clinical attachment loss and no radiographic evidence of alveolar bone loss. Periodontal surgery was performed either to remove diseased tissue (granulation tissue from alveolar defects) in patients with periodontal disease or to remove healthy tissue for cosmetic purposes such as crown lengthening, gingival thinning and cosmetic grafting in healthy individuals.

### 2.3. Isolation of Peripheral and Oral Blood Mononuclear Cells

Written informed consent approved by the UCLA Institutional Review Board (IRB# 11-000781-CR00010; Study ID#11-00781; Committee: UCLA Medical IRB 2) was obtained from healthy individuals and periodontitis patients, and all procedures were approved by the UCLA-IRB. Peripheral blood mononuclear cells (PBMCs) were isolated from peripheral blood as described before [22]. To obtain oral-gingival mononuclear cells approximately 3–6 mL of oral blood was drawn using 6 mL syringe with 16 G needle containing 0.5 mL of heparin. Oral blood was obtained during flap surgery and from granulation tissue (diseased tissue) around alveolar defects or from supra-periosteal tissue (healthy tissue). Collected oral blood was then added to 1:1 ratio of 1 × PBS and layered slowly on a ficoll gradient solution. The samples were then centrifuged for 20 min at 2000 rpm. The collected mononuclear cells (oral blood mononuclear cells, OBMCs) were washed twice with 1 × PBS and re-suspended in RPMI with 10% FBS. The cells were then counted using a hemocytometer and the viability was determined using trypan blue and propidium iodide staining and subsequent analysis by microscopy and flow-cytometry, respectively. Peripheral blood was obtained immediately after the recovery of oral blood from the same individuals. Blood and gingival samples were obtained from both male and female donors. The age range for patients with periodontal disease was 29–68 years, and for healthy individuals it was 27–46 years.

### 2.4. Mononuclear Cells Purified from Gingival Tissues

Gingival biopsies were thoroughly washed with 1 × PBS twice and cut into approximately 1 mm pieces. The cut tissues were placed in RPMI supplemented with DNAse (0.15 mg/mL) and collagenase type II (0.59 mg/mL) and incubated on a shaker for 1 h in °C. After that, the released cells in the supernatants were filtered through a 45–60-micron nylon mesh and the cells were collected in a 50 mL conical tube. The remaining undigested tissue was retreated with RPMI in the presence of DNAse and collagenase type II for a second digestion and incubated for an additional hour. The collected cells were layered on a ficoll gradient to separate the lymphocytes. The lymphocytes were then washed twice with 1 × PBS and re-suspended in RPMI with 10% FBS. The cells were then counted and the viability were determined as described above by trypan blue and propidium iodide staining immediately after purification, and after an overnight incubation at 37 °C.

### 2.5. Enzyme-Linked Immunosorbent Assays (ELISAs) and Multiplex Assays

Single ELISAs were performed as previously described [22]. To analyze and obtain the cytokine and chemokine concentration, a standard curve was generated by either two- or three-fold dilutions of recombinant cytokines provided by the manufacturer. For multiple cytokine array, the levels of cytokines and chemokines were examined by multiplex assay, which was conducted as described in the manufacturer’s protocol for each specified kit. Analysis was performed using a Luminex multiplex instrument (MAGPIX, Millipore, Billerica, MA, USA), and data were analyzed using the proprietary software (xPONENT 4.2, Millipore, Billerica, MA, USA).

### 2.6. Cytotoxicity Assays

The ^51^Cr release assay was performed as described previously [23]. Briefly, different numbers of effector cells were incubated with ^51^Cr–labeled target cells. After a 4-h incubation period, the supernatants were harvested from each sample and the released radioactivity was counted using the gamma counter. The percentage specific cytotoxicity was calculated as follows:$$\mathrm{cytotoxicity}=\frac{\mathrm{Experimental}\;\mathrm{cpm}-\mathrm{spontaneous}\;\mathrm{cpm}}{\mathrm{Total}\;\mathrm{cpm}-\mathrm{spontaneous}\;\mathrm{cpm}}$$

LU 30/10^6^ is calculated by using the inverse of the number of effector cells needed to lyse 30% of tumor target cells × 100.

Cytotoxicity was also performed using xCELLigence Real Time Cell Analysis (RTCA). Tumor cells were added to microplates (E-Plates) overnight, before the addition of effector cells at 1:1 effector to target ratios, and the impedance was read by the instrument at different time intervals. Procedure was conducted as described in the manufacturer’s protocol for xCELLigence immunotherapy kit. 

### 2.7. Surface Staining and Cell Death Assays

Staining was performed by labeling the cells with antibodies or propidium iodide (PI), as described previously [22,24,25]. For surface staining, the cells were washed twice using ice-cold PBS + 1%BSA. Predetermined optimal concentrations of specific human monoclonal antibodies were added to 1 × 10^4^ cells in 50 µL of cold PBS + 1%BSA, and were incubated on ice for 30 min. Thereafter cells were washed in cold PBS + 1%BSA and brought to 500 µL with PBS + 1%BSA. Flow cytometric analysis was performed using Beckman Coulter Epics XL cytometer (Brea, CA, USA).

### 2.8. Purification of Human NK Cells

Briefly, PBMCs were obtained after Ficoll-hypaque centrifugation and were used to isolate NK cells using the EasySep^®^ Human NK cell purchased from Stem Cell Technologies (Vancouver, BC, Canada). Isolated NK cells stained with anti-CD16 to measure the cell purity using flow cytometric analysis.

### 2.9. Retroviral Transduction, Transfection and the Generation of Tumor Cell Transfectants

Cells were infected with culture supernatants of NIH 3T3 packaging cells infected with either GFP expressing a transdominant negative allele of IκB [26] or GFP alone. The mutant IκB-alpha (IκBαM) cDNA was excised from pCMX by digesting with *Bam*HI and EcoRV, and cloned into the pMX-IRES-EGFP retroviral vector and cut with *Not*I (Klenow-filled) and *Bam*HI. Forty-eight hours after infection the cells were sorted and high GFP expressing cells were grown and used in the experiments. The generation of tumor cell transfectants was described previously [23,27]. The stability of IkB(S32AS36A) super suppressor transfected cells in blocking NFkB function were regularly checked by western blot analysis and EMSA using nuclear extracts prepared from the cell transfectants, and a luciferase reporter assay described below.

### 2.10. Luciferase Reporter Assay

Cells were plated and maintained in RPMI supplemented with 10% FBS and 1% penicillin/streptomycin before transfection. Transfections were done using an NF-κB Luciferase reporter vector [28] and Lipofectamine 2000 reagent (Invitrogen, Carlsbad, CA, USA) in Opti-MEM media (Invitrogen, CA) for 18 h after which they were treated with TNF-α. The cells were then lysed with lysis buffer and the relative Luciferase activity was measured using the Luciferase assay reagent kit obtained from Promega (Madison, WI, USA).

### 2.11. Fusobacterium nucleatum Preparation

Viable or 1% paraformaldehyde fixed *F. nucleatum* were used for co-cultures with the immune cells and epithelial tumors at 30:2:1; bacteria: PBMCs: HEp2 tumor ratios. Similar results were obtained with either viable or paraformaldehyde fixed bacterial co-cultures with immune cells and epithelial tumors.

### 2.12. Statistical Analysis

An unpaired or paired, two-tailed Student’s *t*-test was performed for experiments with two groups. One-way ANOVA with a Bonferroni post-test was used to compare different groups for experiments with more than two groups. Duplicate or triplicate samples were used in the in vitro studies for assessment. The following symbols represent the levels of statistical significance within each analysis: *** (*p* value < 0.001), ** (*p* value 0.001–0.01), * (*p* value 0.01–0.05).

## 3. Results

### 3.1. Periodontitis Patients’ Oral Blood Exhibited Higher Percentages of NK Cells and Lower Percentages of B Cells in Comparison to Their Peripheral Blood

We first investigated the percentages of different immune cell subsets in oral and peripheral blood of healthy individuals and periodontitis patients. Similar percentages of NK, T and B cells in peripheral and oral blood were found for healthy individuals (Table 1 (upper two rows)). In contrast, decreased percentages of B cells, increased percentages of NK cells, and similar percentages of T cells were observed in oral blood compared to peripheral blood of periodontitis patients (Table 1 (lower two rows) and Appendix A). Overall, these results indicated that oral blood obtained from periodontitis patients contained higher numbers of NK cells when compared to those obtained from their peripheral blood. The percentages of NK cells within the oral blood were similar between heathy and periodontitis patients.

### 3.2. Oral Gingival-Derived Immune Cells from Periodontitis Patients Exhibited Higher Percentages of B Cells and Lower Percentages of T Cells as Compared to Those from Healthy Individuals

We determined the percentages of immune cell subsets in the oral gingival cells using flow cytometric analysis. Higher percentages of CD19 expressing B cells, and lower percentages of CD3 and CD3 + CD4+ expressing T cells were observed in immune cells derived from gingival tissue of periodontitis patients when compared to those of healthy individuals (Table 2). No significant differences were seen for the percentages of CD16 expressing NK cells or CD3 + CD8+ expressing T cells (Table 2). The majority of CD45+ immune cells in gingival tissues were at an activated state as indicated by higher expression of CD69 surface receptor (Appendix A). Gingival tissue derived immune cells from patients also expressed CD28 and CD95 (Fas) surface receptors (Appendix A).

### 3.3. NK Cells and CD8 + T Cells in Oral Blood of Periodontitis Patients Exhibited Significant Levels of Activation

Oral blood mononuclear cells (OBMCs), and peripheral blood mononuclear cells (PBMCs) of periodontitis patients and healthy individuals were left untreated, or treated with IL-2 or PMA plus ionomycin (PMA/I) overnight. To determine the levels of activation, the early activation antigen expression, CD69 was analyzed on the surfaces of CD16 + NK and CD8 + T cells. The surface expression levels of CD69 activation antigens were found to be elevated on the CD16 + NK cells and CD8 + T cells in OBMCs when compared to PBMCs of periodontitis patients (Figure 1). Moreover, increased surface expression levels of CD69 on NK cells were only observed on cells obtained from periodontitis patients and not form the healthy individuals (unpublished material). When CD16 + NK and CD8+ T cells were treated with IL-2 and PMA/I, higher intensity of CD69 was detected both in OBMCs and PBMCs of periodontitis patients, although the intensity remained higher in OBMCs (Figure 1 and Appendix A). PMA/I treatment resulted in a similar or slightly higher intensity of CD69 surface expression on CD16 + NK and CD8 + T cells in OBMCs as compared to PBMCs (Appendix A). Overall, these results indicated that OBMCs obtained from periodontitis patients exhibited higher activation levels.

### 3.4. PBMCs but Not OBMCs Obtained from Periodontitis Patients Demonstrated Significantly Higher Levels of Cell Death When Compared to Those from Healthy Individuals

Activation by PMA/I resulted in an increased level of cell death both in healthy individuals’ and periodontitis patients’ PBMCs but not in OBMCs (Figure 2). Approximately two-fold more cell deaths could be seen in PMA/I-treated PBMCs of periodontitis patients when compared to those from healthy individuals (Figure 2). In contrast, treatment with PMA/I did not exhibit significant cell death in OBMCs either from patients or healthy individuals (Figure 2).

### 3.5. Increased TNF-α Secretion Was Observed from OBMCs of Periodontitis Patients When Compared to Those from Healthy Individuals

OBMCs and PBMCs of periodontitis patients were left untreated or treated with IL-2 or IFN-γ or PMA/I, and the levels of TNF-α secretion were determined after 12 to 18 h. Higher secretion of TNF-α was observed from OBMCs when compared to PBMCs of periodontitis patients (Figure 3A). IFN-γ treatment increased TNF-α secretion moderately by patient OBMCs whereas it induced significant levels of secretion by the patient PBMCs (Figure 3A). Although there was a moderate increase in TNF-α release by IFN-γ treated OBMCs obtained from patient and healthy donors, the increase was not statistically significant (Figure 3A). The lack of increase in TNF-α by IFN-γ could be due to higher basal activation of OBMCs in patients. The levels of TNF-α secretion between PMA/I-treated PBMCs and OBMCs plateaued since PMA/I activates lymphocytes in both cell populations maximally (Figure 3A). Next, we compared the TNF-α secretion in OBMCs obtained from healthy individuals and periodontitis patients. We observed increased secretion of TNF-α in OBMCs from periodontitis patients with and without IL-2 or IFN-γ treatment (Figure 3B). These results demonstrated the increased functional activities of mononuclear cells in oral blood as compared to peripheral blood of periodontitis patients. Furthermore, periodontal disease found to be associated with increased TNF-α secretion in oral blood immune cells.

### 3.6. OBMC-Derived Supernatants from Periodontitis Patients Regulate Secretion of Cytokines by Allogeneic Healthy PBMCs

Since chronic periodontitis was shown to be associated with an altered balance between anti-inflammatory and pro-inflammatory cytokines, we tested the ability of OBMCs-derived supernatants from periodontitis patients and healthy individuals to modulate IFN-γ, IL-6, TNF-α and IL-1β secretions in untreated and IL-2-treated allogeneic PBMCs isolated from healthy individuals. When compared to OBMC-derived supernatants from healthy individuals, addition of those derived from OBMCs of periodontitis patients to IL-2 activated PBMCs had lower priming/activating capability of PBMCs to secrete IFN-γ and TNF-α, whereas IL-6 induction was similar between the two, likely due to the plateauing effect (Figure 4A–C). IL-1β induction was also lower in the presence of OBMC-derived supernatants from periodontitis patients, but with no statistical differences (Figure 4D). When adding to untreated PBMCs, OBMC-derived supernatants from patients or healthy individuals exhibited no change in IFN-γ induction whereas much higher levels of IL-6, TNF-α and IL-1β were induced in those treated with OBMC-derived supernatants from patients compared to those from healthy individuals (Figure 4A–D).

### 3.7. NFkB Deletion in Oral Epithelial Cells Increases IFN-γ Secretion by PBMCs and NK Cells in the Presence or Absence of Fusobacterium nucleatum

Since immune effectors in the periodontal tissues interact with a number of stromal cells as well as with the epithelial cells to induce cytokine secretion, we determined the induction of cytokines in the presence of a number of either non modified or genetically modified oral epithelial cell lines as shown below in the presence and absence of *F. nucleatum*. First, we determined NK cell-mediated cytotoxicity against the pRcCMV vector alone (HEp2-pRcCMV), or IκB(S32AS36A) transfected HEp2 (HEp2-IκB(S32AS36A)) cells as targets. The NK cell-mediated cytotoxicity was higher against HEp2-IκB(S32AS36A) cells in comparison to HEp2-pRcCMV cells by untreated, IL-2-treated or anti-CD16 mAbs treated NK cells (Figure 5A–C). Similar results were seen in oral squamous carcinoma cells (OSCCs) and human oral keratinocytes (HOK) cells. Inhibition of NFkB by the IkB(S32AS36A) super-repressor retroviral vector was confirmed by measuring NFkB activity using a luciferase reporter assay (Appendix A). Overall, the NK cell-mediated cytotoxicity was higher against NFkB knock-down OSCCs and HOK cells when compared to EGFP transfected cells (Appendix A).

To determine the effect of immune cells’ interaction with epithelial cells in the presence and absence of *F. nucleatum*, we left PBMCs and NK cells untreated or treated them with IL-2 before they were co-cultured with HEp2-IκB(S32AS36A) and HEp2-pRcCMV in the presence or absence of *F. nucleatum.* We observed increased secretion of IFN-γ in co-cultures of PBMCs with HEp2-IκB(S32AS36A) tumors compared to those with HEp2-pRcCMV in the presence or absence of *F. nucleatum* (Figure 6A). Although similar results to IFN-γ were seen for TNF-α, GM-CSF, IL-13, MCP-1, and RANTES, the secretion of IL-6 was higher in immune cell cultures with HEp2-pRcCMV in comparison to those with HEp2-IκB(S32AS36A) tumors (Appendix A). Increased IFN-γ secretion was also observed when untreated and IL-2-treated NK cells were co-cultured with HEp2-IκB(S32AS36A) tumors compared to HEp2-pRcCMV in the absence or presence of *F. nucleatum* (Figure 6B). Similar results were seen when IL-2 treated NK cells were co-cultured with NFkB knock-down OSCCs and HOK cells in comparison to those cultured with EGFP transfected cells (Appendix A). The treatment with *F. nucleatum* increased IFN-γ in all culture conditions and, the highest was seen in IL-2-treated PBMCs or NK cells co-cultured with HEp2-IκB(S32AS36A) tumors (Figure 6A,B). Increased secretion of TNF-α, IL-8, GM-CSF, and IL-13 was also seen when NK cells were co-cultured with HEp2-IκB(S32AS36A) in comparison to HEp2-pRcCMV tumors in the absence of *F. nucleatum* (Appendix A). The secretion of IL-6 was higher in HEp2-pRcCMV in comparison to HEp2-IκB(S32AS36A) tumors (Appendix A). Furthermore, IL-6 secretion was lower when NK cells were co-cultured with NFkB knock-down OSCCs and HOK cells in comparison to those cultured with EGFP transfected cells (Appendix A). Interestingly, IL-13 and GM-CSF secretions were inhibited by *F. nucleatum* in IL-2-treated NK cells co-cultured with HEp2-IκB(S32AS36A) and HEp2-pRcCMV cells (Appendix A). Overall, these results demonstrated that NFkB knock-down cells in comparison to their non-knock-down counterparts are more active inducers of PBMCs and NK cells to secrete IFN-γ, and presence of F. *nucleatum* enhances this functional activation.

## 4. Discussion

Although ample progress has previously been made regarding periodontal disease, a clear understanding of immune interaction with neighboring stromal cells with probable genetic and epigenetic modifications in the presence of complex oral microorganisms is still limited. In previous studies disease-relevant assumptions or interpretations were usually made based on the interaction of periodontal pathogens with peripheral blood immune effectors. In addition, the use of tissue-associated immune effectors from the infected sites also provided limited information regarding the disease pathogenesis since these cells were isolated largely at a non-functional state due to the harsh recovery methods. Indeed, in our experience the majority of the immune effectors isolated from the gingival tissues underwent cell death when the cells were incubated greater than 3–4 h. Furthermore, they were unable to secrete cytokines or mediate cytotoxicity even though they exhibited increased levels of activation markers on their surface. Thus, such experiments are great to assess the phenotype of the immune effectors at the time of isolation, but lack ability to provide functional readouts. These observations, therefore, prompted us to design strategies to obtain immune cells from the oral site using oral blood which remained functional and had the capacity to respond when exposed to self-bacterial and viral microflora. Therefore, to understand potential similarities and differences between peripheral, oral and tissue-derived immune effectors, initially we obtained and characterized immune subsets from each of these sites in healthy individuals and from the patients with periodontal disease (Table 1 and Table 2).

The rationale for using orally-derived blood to compare with the peripheral blood and tissue-derived immune effectors is; (1) to observe whether there were quantitative differences between the immune subsets from different compartments, and (2) to determine and compare the levels of their functional competency. It is likely that orally-derived immune effectors contain elements of saliva in addition to oral microorganisms, and thus exhibit phenotypic and functional properties that are distinct from those obtained from peripheral blood or even those found in the gingival tissues. Nevertheless, the analysis could shed light on their differences and help identify important questions regarding the pathogenesis of periodontal disease. For example, are there any phenotypic similarities between the oral blood immune effectors and those recovered from the gingival tissues? When comparing CD69 expression we observed similar increase in CD69 surface expression in oral blood mononuclear cells compared to those extracted from gingival tissues. It is interesting to note that lymphocytes obtained from healthy gingival tissues also expressed higher levels of CD69 expression when compared to the patients with periodontitis. However, the numbers of immune effectors obtained from healthy gingival tissues were far less than those obtained from patients with periodontitis.

The increased activation of orally-derived mononuclear cells from patients could be due to their exposure to periodontal pathogens during the recovery method. In this regard the oral mononuclear cells from patients may recognize and become strongly activated to their oral flora and/or they may be at a stage of maturation which is more susceptible to activation signals delivered by the oral pathogens. Indeed, it is believed that the nature of host immune cells is the determining factor for the response to periodontal pathogens [29,30]. In addition, patients with periodontal disease are shown to have genetic predisposition for increased cytokine secretion, particularly IL-1β and TNF-α [31]. Therefore, the studies reported here are likely to shed some light on the mechanisms of immune cell activation in patients since both the immune effectors and the combination of pathogenic oral bacterial and viral flora are derived from the same oral niche.

Orally-derived immune cells from patients with periodontal disease exhibited an increase in the activation of NK and CD8+ T cell fractions after an overnight incubation, and secreted higher levels of TNF-α, whereas those obtained from healthy donors exhibited a profile closer to those obtained from naïve peripheral blood immune cells. In this regard we have recently reported that activated NK cells are able to activate CD8+ T cells [32]. As indicated, the difference could be due to either higher innate capacity of the immune effectors to become activated and/or the increased activating capacity of periodontal pathogens in patients compared to healthy controls. Our results also indicated that mononuclear cells obtained from the oral blood did not undergo significant activation induced cell death after treatment with PMA/I when compared to peripheral blood, even though OBMCs demonstrated increased levels of activation when compared to PBMCs. No significant differences could be observed in cell death between OBMCs obtained from healthy individuals and those with periodontitis in the presence and absence of PMA/I treatment. The resistance of oral mononuclear cells to apoptotic cell death has been documented previously in chronic periodontal lesions [33,34]. Both IL-4 and Fas Ligand were found to be decreased or not expressed within the gingival tissues even though the IL-4 and Fas receptors were expressed on the immune cells, suggesting a lack of ligand for decreased cell death [33,34]. However, since PMA/I treatment bypasses the receptor mediated effects, this implies other underlying mechanisms for the lack of cell death in OBMCs. Whether there are pressures for selecting longer surviving immune cells in OBMCs requires further investigations. Therefore, continued inflammation and tissue damage in periodontal disease may partly be mediated by longer surviving oral mononuclear cells.

Activation-induced cell death in immune effectors is a well characterized cellular outcome which occurs upon potent activation of immune cells. As indicated above, PBMCs treated with PMA/I underwent significant cell death, whereas OBMCs demonstrated considerably lower levels of cell death upon activation with PMA/I. No significant correlation could be seen between levels of activation (using CD69 expression) and the extent of cell death in immune effectors isolated from peripheral and oral blood. Indeed, more cell death was observed in PBMCs that had a slightly lower CD69 expression when compared to OBMCs after treatment with PMA/I (Figure 2 and Appendix A). Therefore, other factors in addition to the extent of activation may be responsible for continued survival of these cells in an oral microenvironment. Whether components of saliva in addition to signals delivered by bacterial and viral flora provide a survival advantage for mononuclear cells awaits future investigation.

Higher numbers of CD16+ NK cells were observed in the oral blood of patients with periodontal disease when compared to the peripheral blood and gingival tissues. NK cells are a subset of lymphocytes which mediate first line defense against a variety of tumors and microorganisms. Indeed, NK cells are one of the primary sources of secreted IFN-γ. Thus, activation of NK cell function and elaboration of cytokines such as IFN-γ is essential for the expansion and amplification of immune responses mediated by the effectors of both an innate and adaptive immune system. In addition, IFN-γ is important in differentiation of stromal cells and subsequent cessation of NK cell function since NK cells are known to become activated by the stem cells and not by the well-differentiated cells [35,36]. Thus, IFN-γ can regulate its own production negatively and limit the levels of inflammation [35,36,37,38].

The rationale for a lower percentage of peripheral blood NK cells compared to oral blood NK cells in patients with periodontal disease is not clearly known yet. One can speculate regarding the increased homing of these cells to oral tissues from the peripheral blood in which a higher gradient of chemokines is secreted by epithelial cells upon activation by pathogenic organisms. In addition, it is also possible that lower percentages of NK cells seen in the gingiva of patients compared to those in the oral blood could be due to the higher induction of activation-induced cell death in NK cells in the gingiva where higher activation signals are given to the NK cells by the gingival microenvironment, a phenomenon that we and others have established previously in NK cells [39]. As for the differences in percentages of B cells in the peripheral and oral blood and gingival tissue, the decrease in the oral blood seems to be compensated for by the increase in the percentages of B cells in the gingival tissues of patients with periodontal disease. Such differences in the quantity of B cells could not be seen in different tissue compartments in healthy individuals. The percentages of T cells remained high in all compartments in healthy individuals as well as in patients with periodontal disease.

When added to IL-2-activated allogeneic PBMCs, supernatants obtained from OBMCs of periodontitis patients were only able to increase IFN-γ secretion by about 2.4-fold, whereas those obtained from healthy individuals increased by 5.4-fold. Since supernatants obtained from OBMCs of periodontitis patients contain significant levels of pro-inflammatory cytokines such as TNF-α, IL-6 and IL-1β, these cytokines may be inhibitory for the induction of IFN-γ. Indeed, we have previously shown that both TNF-α and IL-6 inhibit secretion of IFN-γ by the NK cells [24,27]. Higher induction of pro and anti-inflammatory cytokines in the presence of decreased IFN-γ secretion may, therefore, tilt the balance towards the establishment of chronic inflammation in periodontal infections. Indeed, we have shown in a series of studies that IFN-γ is important in the differentiation of stromal cells and the limitation of inflammation [36,40,41]. In addition, healthy stem cells or stem-like cancer stem cells, but not differentiated cells, are able to activate NK cells due to a lack of or decreased surface expression of MHC-class I [36,42,43]. Therefore, it is possible that due to a lower release of IFN-γ during chronic inflammation, adequate induction of MHC-class I expression does not occur in the stromal cells, thereby leading to a decreased activation and release of IFN-γ by the NK cells which maintains the stromal cells at a lower level of differentiation resulting in the chronicity of inflammation. These possibilities are currently under investigation in our laboratory.

Since both bacterial and viral agents have been implicated in the pathogenesis of periodontal disease [44,45,46,47], it is likely that suboptimal induction of IFN-γ will have profound effects on the clearance of both types of organism and the resolution of inflammation [48,49,50]. Therefore, the increase in the numbers and activation of NK cells in periodontal infection may have a direct relationship to the increase in colonization with the viral and bacterial infections since NK cells are the primary effectors that mediate lysis of virally infected cells. In addition, by secreting IFN-γ, NK cells will be able to increase lysis of bacteria by augmenting the activation of monocyte-macrophages and dendritic cells [51].

It is known that the number of bacteria in the periodontal tissues of the patients does not correlate with the severity of periodontal disease, and that it is likely that other factors are involved in the pathogenesis of disease [1,52]. Thus, periodontal disease is multifactorial. Since periodontal disease has been shown to have genetic as well as environmental components, it is likely that mutations either in the stromal cells or in the immune cells could likely exacerbate the disease [52,53,54,55,56]. We have shown previously that many genetic alterations in particular deletions of important genes in stromal cells or in the immune cells can directly activate NK cells which could lead to the expansion and differentiation of T cells [32,57,58]. In order to determine whether such interactions may have a significant activating effect on PBMCs and NK cells in the presence of an oral pathogenic bacteria *F. nucleatum* (known to be associated with periodontal disease), we determined the effect of NFkB knock-down epithelial cells in the activation of PBMCs and NK cells in the presence of *F. nucleatum*. Significant activation of PBMCs and NK cells and the augmented release of different cytokines and chemokines were observed under such conditions (Figure 6, Appendix A and Appendix A). Indeed, Jung et al. and Raje et al. reported previously how Nuclear factor-kappa B Essential Modulator (NEMO) deficiency might lead to a genetic predisposition named “Mendelian Susceptibility to Mycobacterial Disease (MSMD)”, which increases susceptibility to mycobacterial infections [59,60]. In addition, Javali et al. indicated that periodontal disease was the initial oral manifestations of abdominal tuberculosis that was caused by mycobacteria infections, implicating this organism in periodontal disease [61]. Therefore, since NEMO deficiency may lead to an overall manifestation of immune deficiency through poor response to bacterial and fungal infections from NK, B and T-cells, as well as neutrophils, macrophages and dendritic cells, the bacterial infections from Gram-negative anaerobic bacteria, which are implicated in the periodontal diseases, are also likely. It is of interest to note that NFkB deletion in epithelial tumors increased IFN-γ secretion in the presence of decreased IL-6 in the co-cultures of these tumors with immune cells (Figure 6 and Appendix A). Such observation is likely due to the significantly decreased secretion of IL-6 in NFkB knock-down tumors as well as decreased synergistic induction of IL-6 in the co-cultures of immune cells with NFkB knock-down tumors (Appendix A). Increased activation of NK cells and higher lysis of the tumors by the NK cells might also contribute to the decreased IL-6 secretion seen in the co-cultures of NFkB tumors with the NK cells. These findings indicate that cellular alternations in those that interact with the immune cells and/or in immune cells can exacerbate the immune function in the presence of periodontal pathogens, and could be the likely cause of the destruction which is seen in periodontitis patients. Although NFkB deficiency in epithelial cells or in other stromal cells has not been reported to be one of the major causes of periodontal disease, nevertheless, such findings may implicate the potential for genetic alterations in the exacerbation of periodontal disease.

Overall, our studies have identified a number of characteristics of immune effectors from different tissue compartments, which may have relevance to the pathogenesis of periodontal infections.

## Figures and Tables

**Figure 1 jcm-10-00875-f001:**
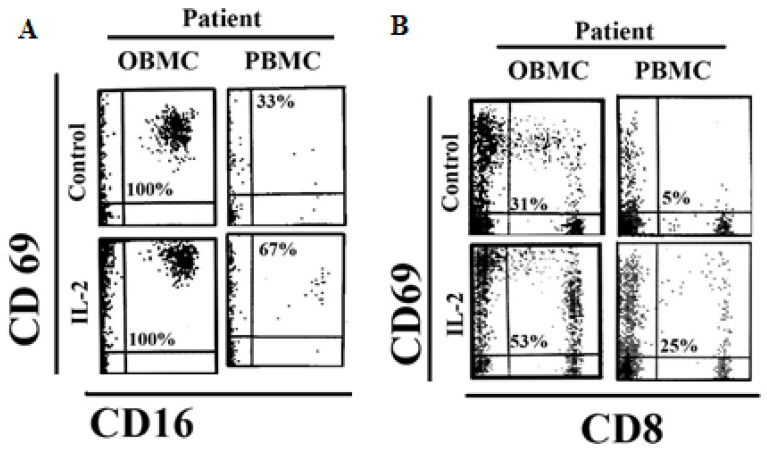
Increased activation of NK and CD8 + T cells in oral blood when compared to peripheral blood of periodontitis patients. Mononuclear cells (1 × 10^6^/mL) obtained from oral and peripheral blood of periodontitis patients left untreated and treated with IL-2 (500 U/mL) for 18–24 h after which they were washed twice. The surface expression levels of CD16 and CD69 (**A**), and CD8 and CD69 (**B**) were determined using flow cytometry. IgG2 isotype was used as controls. Numbers in each quadrant represent the percentages of stained sub-population for the specific antigen. The oral and peripheral blood was obtained from the same donors. One of four representative experiments is shown in this figure.

**Figure 2 jcm-10-00875-f002:**
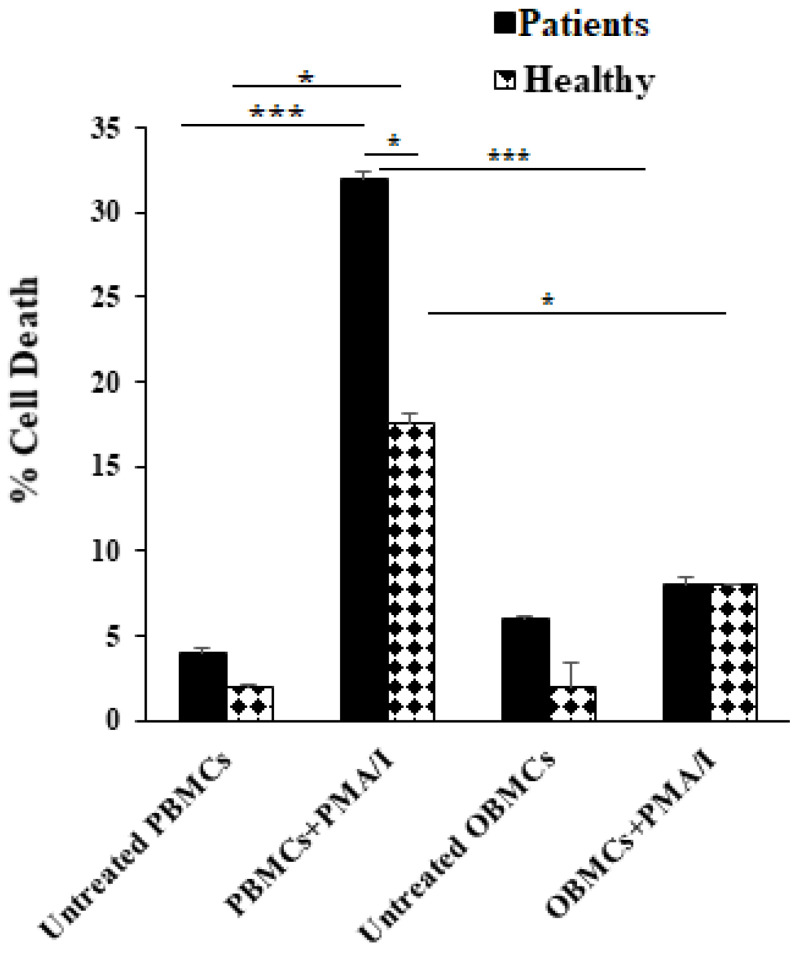
Higher levels of cell death were observed in periodontitis patients’ PBMCs when compared to those from healthy individuals. Mononuclear cells obtained from peripheral and oral blood of healthy individuals and periodontitis patients were treated with PMA (10 ng/mL) + ionomycin (10 ng/mL) for 10–14 h. The samples were then washed and propidium iodide (30 µg/mL) was added to each sample. The percentages of dead cells were determined by flow cytometric analysis. PBMCs and OBMCs were obtained from the same donors. One of four representative experiments is shown in this figure. *** (*p* value <0.001), * (*p* value 0.01–0.05).

**Figure 3 jcm-10-00875-f003:**
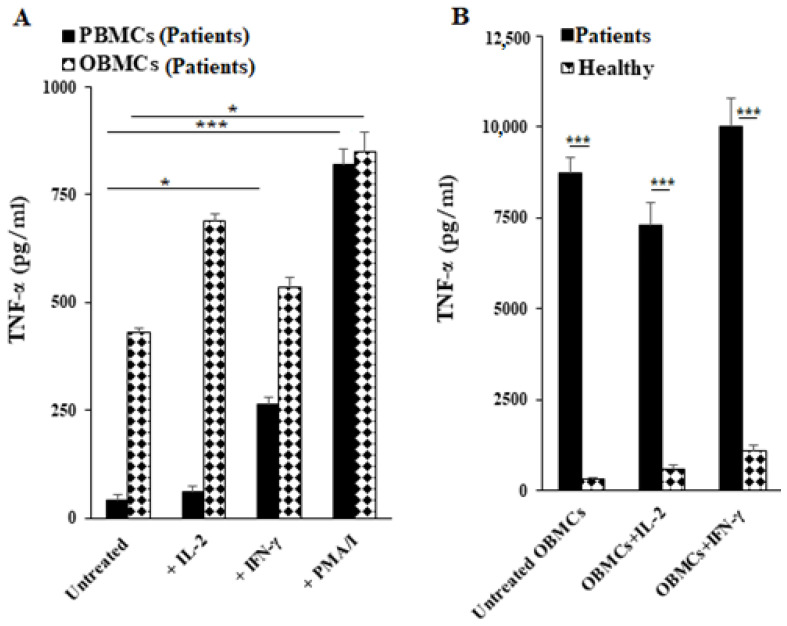
Periodontitis patients’ OBMCs secreted higher levels of TNF-α secretion in comparison to their PBMCs; Periodontitis patients’ OBMCs secreted higher levels of TNF-α when compared to those from healthy individuals. PBMCs and OBMCs of periodontitis patients were treated with IL-2 (500 U/mL), IFN-γ (500 U/mL) and PMA (10 ng/mL) + ionomycin (10 ng/mL). After 12–18 h of incubation, supernatants were harvested and the levels of TNF-α secretion were determined using single ELISA. *** (*p* value < 0.001), * (*p* value 0.01–0.05), *p* values were obtained for differences between untreated or IL-2 or IFN-γ treated PBMCs and OBMCs obtained from the periodontitis patients (**A**). OBMCs obtained from healthy individuals and periodontitis patients were left untreated, treated with IL-2 (500 U/mL) or treated with IFN-γ (500 U/mL). After an overnight incubation the supernatants were harvested and subjected to a single ELISA to determine TNF-α secretion levels. *** (*p* value < 0.001), *p* value was obtained for the difference between TNF-α secretion by OBMCs obtained from healthy individuals and periodontitis patients; control, IL-2 and IFN-γ treated OBMCs (**B**). One of four representative experiments is shown in this figure.

**Figure 4 jcm-10-00875-f004:**
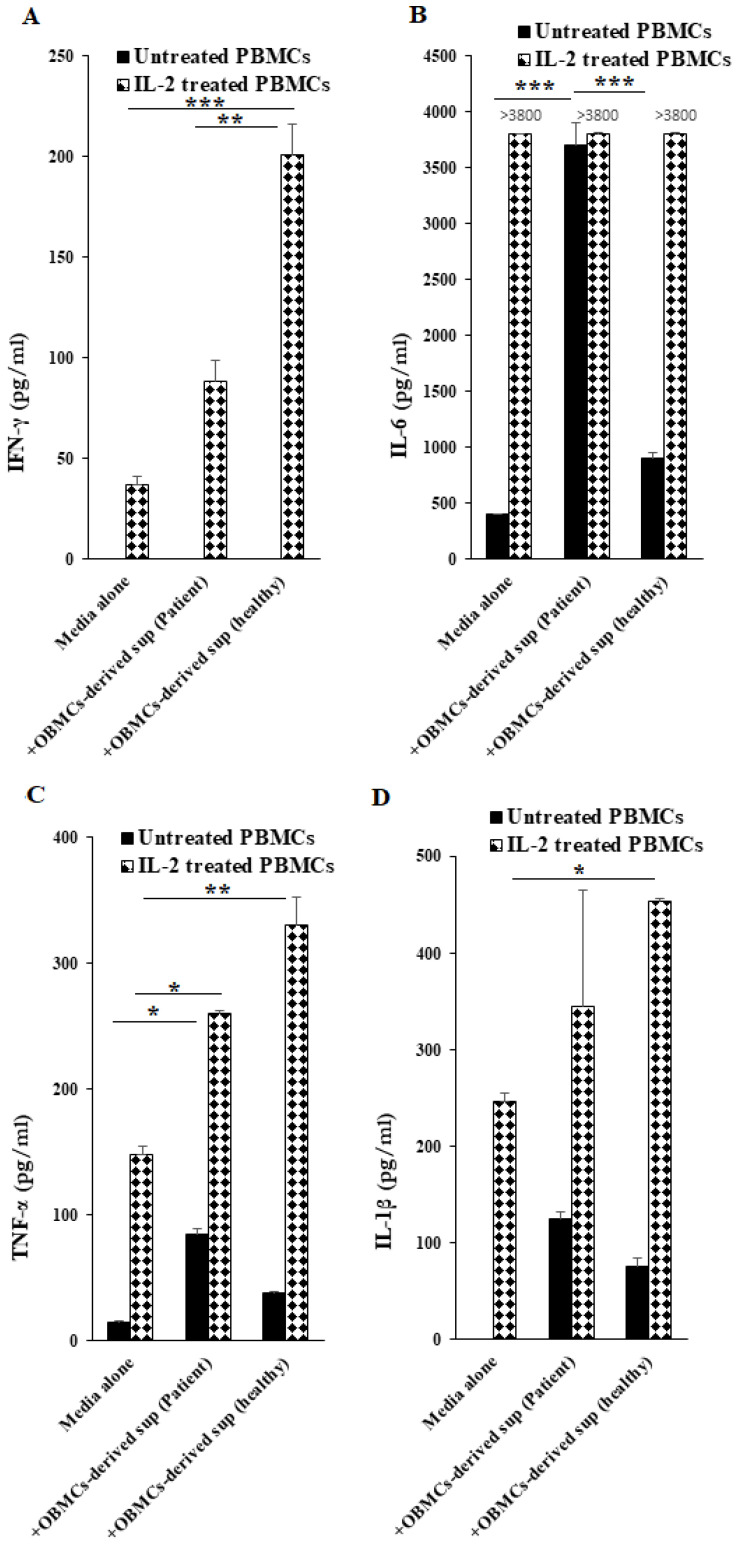
Periodontitis patients’ OBMCs-derived supernatant had decreased ability to activate allogeneic healthy PBMCs in comparison to healthy individuals’ OBMC-derived supernatant. Untreated OBMCs (1 × 10^6^/mL) of healthy individuals and periodontitis patients were incubated at 37 °C for 6–12 h before supernatants were harvested. Allogeneic healthy PBMCs (1 × 10^6^/mL) were left untreated or treated with IL-2 (500 U/mL) for 12–18 h, after which, OBMCs-derived supernatants were added to PBMCs, and 18–20 h later the supernatants were harvested to determine IFN-γ (**A**), IL-6 (**B**), TNF-α (**C**), IL-1β (**D**) secretion using single ELISAs. One of four representative experiments is shown in this figure. *** (*p* value < 0.001), ** (*p* value 0.001–0.01), * (*p* value 0.01–0.05).

**Figure 5 jcm-10-00875-f005:**
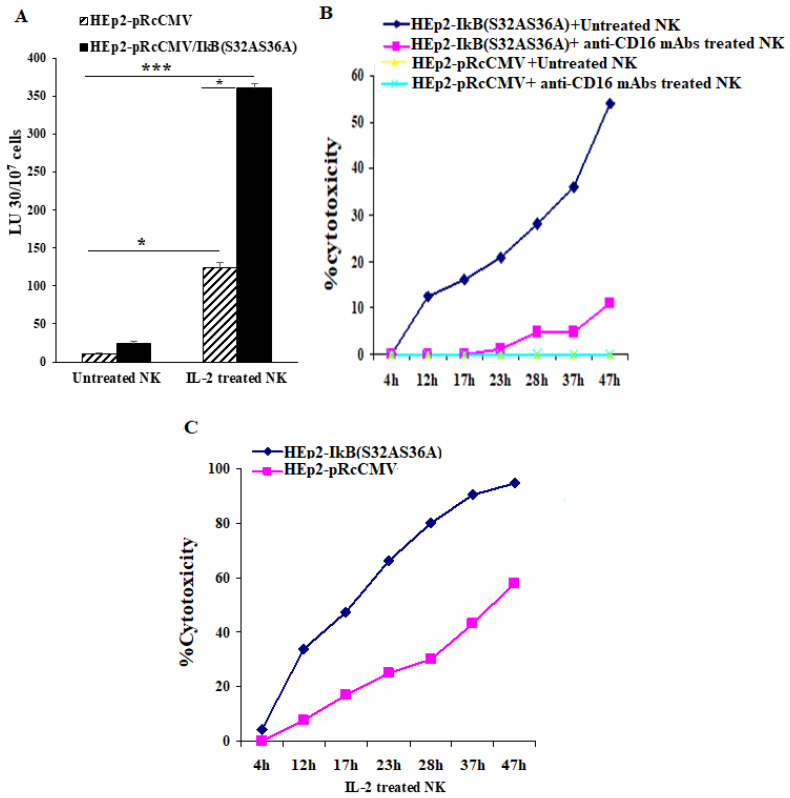
NFκB blocking in HEp2 tumors increased their susceptibility to NK cell-mediated cytotoxicity. NK cells were left untreated and treated with IL-2 (500 U/mL) overnight before they were added to ^51^Cr-labeled pRcCMV vector alone or IκB(S32AS36A) transfected HEp2 cells at various effector-to-target ratios. NK cell-mediated cytotoxicity was measured using a standard 4-h ^51^Cr release assay. The lytic units (LU) 30/10^7^ cells were determined using the inverse number of NK cells required to lyse 30% of target cells × 100 (**A**). *** (*p* value < 0.001), * (*p* value 0.01–0.05). NK cells were left untreated and treated with anti-CD16 mAbs (3 µg/mL) overnight before they were added to ^51^Cr-labeled pRcCMV vector alone or IκB(S32AS36A) transfected HEp2 cells at various effector-to-target ratios. NK cell-mediated cytotoxicity was measured using xCELLigence Real Time Cell Analysis (RTCA) (**B**). NK cells were treated with IL-2 (500 U/mL) overnight before they were added to ^51^Cr-labeled pRcCMV vector or and IκB(S32AS36A) transfected HEp2 cells at various effector-to-target ratios. NK cell-mediated cytotoxicity was measured using xCELLigence Real Time Cell Analysis (RTCA) (**C**). One of four representative experiments is shown in this figure.

**Figure 6 jcm-10-00875-f006:**
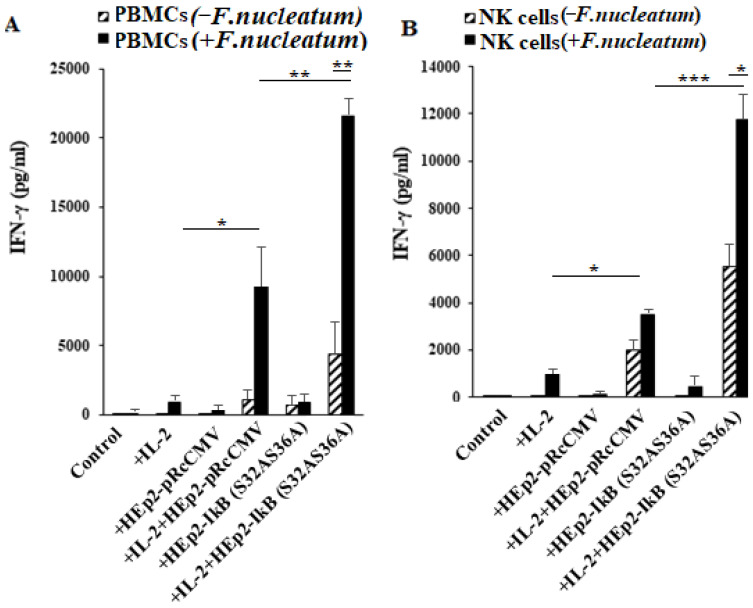
Increased IFN-γ secretion levels were observed in *F. nucleatum* treated PBMCs and NK cells; highest secretion levels were seen in the presence of NFκB blocked HEp2 tumors. PBMCs were left untreated or treated with IL-2 (2000 U/mL) overnight before their co-culture with HEp2-pRcCMV or HEp2-IκB(S32AS36A) cells in the presence or absence of *F. nucleatum* (bacteria: PBMCs: Hep2 tumors at 30:2:1 ratios). After an overnight incubation, the supernatants were harvested to determine IFN-γ secretion using multiplex cytokine array assay, one of four representative experiments are shown in the figure (**A**). NK cells were left untreated or treated with IL-2 (2000 U/mL) overnight before their co-culture with HEp2-pRcCMV or HEp2-IκB(S32AS36A) cells in the presence or absence of *F. nucleatum* at (bacteria: PBMCs: Hep2 tumors at 30:2:1 ratios). After an overnight incubation, the supernatants were harvested to determine IFN-γ secretion using multiplex cytokine array assay (**B**). One of four representative experiments is shown in this figure *** (*p* value < 0.001), ** (*p* value 0.001–0.01), * (*p* value 0.01–0.05).

**Table 1 jcm-10-00875-t001:** Percentages of lymphocyte cell subsets in peripheral and oral blood of healthy individuals and periodontitis patients.

		CD16+	CD3+	CD19+	CD3 + CD4+	CD3 + CD8+
Healthy	Peripheral blood	15.5 ± 3.5 **	74 ± 7	10.5 ± 3.5 *	55 ±7	19 ± 9
	Oral blood	15.8 ± 9.7	74 ± 6.9	7.8 ± 7.4	52 ± 6	22 ± 10
Patients	Peripheral blood	3.5 ± 0.7 **	71 ± 9.9	24.5 ± 9 *	51 ± 3	20 ± 1
	Oral blood	16 ± 10	73 ± 15	6.7 ± 4	53 ± 2	20 ± 7

Mononuclear cells were isolated from peripheral and oral blood of healthy individuals and periodontitis patients using Ficoll-Hypaque density gradient. The percentages of lymphocyte subsets were determined immediately after purification using specific antibody staining followed by flow cytometric analysis. The IgG2 isotype was used as a control. Significant differences were obtained between the levels of CD16 + NK (** *p* value of 0.001–0.01) and CD19 + B cells (* *p* value 0.01–0.05) in the peripheral blood between healthy individuals and those of the periodontitis patients, and for CD16+ NK cells (** *p* value of 0.001–0.01) and CD19 + B cells (* *p* value 0.01–0.05) between peripheral and oral blood in periodontitis patients. No significant differences (*p* = 0.8) were observed in peripheral and oral blood between healthy individuals and periodontitis patients for the levels of CD3, CD4 and CD8 lymphocyte subpopulations. The numbers for each immune subset reflect the mean percentages derived from six donors ± standard deviation.

**Table 2 jcm-10-00875-t002:** Percentages of lymphocyte cell subsets in gingival tissues obtained from healthy individuals and periodontitis patients.

	CD16+	CD3+	CD19+	CD3 + CD4+	CD3 + CD8+
Healthy	4.7 ± 5.0	80 ± 19	6.7 ± 5.9	62 ± 8	18 ± 10
Patients	5.05 ± 3.9	68 ± 23	27 ± 23	47 ± 10	21 ± 4

Gingival tissue-associated mononuclear cells from healthy individuals and periodontitis patients were obtained as described in the Materials and Methods section. The percentages of lymphocyte subsets in mononuclear cells were determined using specific antibody staining followed by flow cytometric analysis. IgG2 isotype was used as controls. *p* = 0.14 was obtained for the difference between the levels of CD19 + B cells obtained from the healthy individuals and periodontitis patients. No significant differences (*p* = 0.7) were observed for CD16, CD3, CD4 and CD8 lymphocyte subpopulations between healthy individuals and periodontitis patients. The numbers for each immune subset reflect the mean percentages derived from six donors ± standard deviation.

## Data Availability

The data presented in this study are available in the article or Appendix A of “Phenotypic and functional alterations of immune effectors in periodontitis; a multifactorial and complex disease of oral gingival tissues”.

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
