# Peer review of "Phenotypic and Functional Alterations of Immune Effectors in Periodontitis; A Multifactorial and Complex Oral Disease"

_jcm, 2021, doi:10.3390/jcm10040875_

Round 1

Reviewer 1 Report

This is a well written and presented paper. I have several questions which to me seem important in assessing the overall meaning of the results as they may indicate the state of progression of the disease in each case and whether the change are related to that. We know the white cell counts will change with the state of the activation of the infectious process so simply looking at the % change would suggest we are missing some important data. 1) did the total white cell counts (blood and oral) vary greatly between and within the two study cohorts? 2) if so did the % distribution of the different cell types correlate with the change in the different total white cell counts? 3) did the different total white cell counts change with the cytokine changes identified? 4) did the fusobacterium cytokine results correlate with these changes? Some vital questions could be answered if this data is presented.

Author Response

This is a well written and presented paper.

We appreciate the kind word of the reviewer and happy that we could provide a well written study.

I have several questions which to me seem important in assessing the overall meaning of the results as they may indicate the state of progression of the disease in each case and whether the change are related to that.

We appreciate the thoughtful and constructive questions of the reviewer, and are providing point by point answers below.

We know the white cell counts will change with the state of the activation of the infectious process so simply looking at the % change would suggest we are missing some important data. 1) did the total white cell counts (blood and oral) vary greatly between and within the two study cohorts?

We appreciate the thoughtful question from the reviewer. Because this point has been investigated previously by several laboratories and WBCs were found to be higher in the majority of chronic periodontitis patients, when compared to control healthy individuals (1-3), we did not undertake detailed analysis for this point in order to avoid criticism regarding the novelty of the data, however, overall, we did observe increased levels of WBC in periodontal patients. Since we were interested in the functional aspects of lymphocytes, we limited the manuscript to assess the function of lymphocytes, which are the novel aspects of this paper.

2) if so did the % distribution of the different cell types correlate with the change in the different total white cell counts?

If I understand the question correctly the reviewer is asking whether the higher WBC count could account for the differences which are noted in terms of percentages of different lymphocyte subsets. Whether somehow the bone marrow of the patients are selectively increases or decreases generation of certain subsets under high turn over of WBCs requires further in vivo studies. It is clear that B cell populations are elevated in the peripheral blood of periodontal patients which also reflects the increase in the gingival tissues, however, the levels in oral blood was decreased. Whether this is reflected on the faster speed of migration from oral blood to gingival tissues requires further investigation. These observations are interesting, and as the reviewer so astutely pointed out, there are many questions that has arisen from these observations, which would need additional studies. The present study is more focused on the function of different lymphocytes in terms of cell death, pro-inflammatory cytokine secretion, and the putative role that immune cells, bacteria and epithelial cells may play in the process.

To understand the point raised by the reviewer fully we need to carry out different in vivo experiments such as labeling different subsets of lymphocytes and tracking them where they migrate in an animal model system. Also important is to understand the dynamics of B, T  and NK cell generation from the bone marrow.  

3) did the different total white cell counts change with the cytokine changes identified?

The proportions of different cell types remain similar in the patients even though the WBC are found to be elevated in the periodontal patients, therefore, the overall changes depend on the increase in the function of the different immune cells than changes in the overall increase in WBC since the same number of cells are used to assess their function from periodontal patient as compared to healthy individuals. Also both NK cells and T cells are the main sources of IFN-g production, therefore, we used purified population of NK cells to establish the functional increase in IFN-g to correlate with those seen with PBMCs.

4) did the fusobacterium cytokine results correlate with these changes? Some vital questions could be answered if this data is presented.

The F. nucleatum data is from bacterial prep in the lab with the immune cells from healthy individuals with two epithelial tumors which were generated in the lab. This is used as a model that could explain why in periodontitis patients we see such elevated levels of pro-inflammatory cytokines as discussed in the discussion section.

  1. Kumar BP, Khaitan T, Ramaswamy P, Sreenivasulu P, Uday G, Velugubantla RG. Association of chronic periodontitis with white blood cell and platelet count - A Case Control Study. J Clin Exp Dent. 2014;6(3):e214-e7.
  2. Al-Rasheed A. Elevation of white blood cells and platelet counts in patients having chronic periodontitis. The Saudi Dental Journal. 2012;24(1):17-21.
  3. Botelho J, Machado V, Hussain SB, Zehra SA, Proença L, Orlandi M, et al. Periodontitis and circulating blood cell profiles: a systematic review and meta-analysis. Exp Hematol. 2021;93:1-13.

Reviewer 2 Report

This is a really interesting study regarding functional alterations of immune cells in oral, and peripheral, blood of  patients with periodontitis compared with healthy controls. However, there are many variables included in this study.

-  Title is confusing because periodontitis affects alveolar bone, not only gingival tissues. So, please omit… a multifactorial and complex disease of oral gingival tissues.

-Define PBMCs in the abstract.

- Although “Periodontitis patients’ oral blood exhibited higher percentages of NK cells… in comparison to their peripheral blood”, it is important to highlight that there is not difference in the CD16+ NK percentage in oral blood  between healthy and periodontal patients. Considering that  there is a difference in the age of periodontal (29-68 years) and healthy patients (27-46 years), could the lower percentage of NK cells in peripheral blood be a matter of aging?.

-There is not difference in the percentage of NK cells  in Oral gingival-derived immune cells. Please argue this in the discussion.

- “Since immune effectors in the periodontal tissues interact with a variety of stromal cells including epithelial cells”. In this way it seems that epithelial cells are stromal cells. Please, rephrase this sentence.

-Considering that “The increased activation of orally derived mononuclear cells from patients could be  due to their exposure to periodontal pathogens….”, this reviewer consider that one of the main points of this paper is the effect of periodontal environment on immune cells. Given that bacteria contribute to the development of periodontal disease, please evaluate the effect of F. nucleatum  coincubation on orally derived immune cells (mononuclear cells) from healthy patients . Determine  pro-inflammatory cytokines levels.

Author Response

This is a really interesting study regarding functional alterations of immune cells in oral, and peripheral, blood of patients with periodontitis compared with healthy controls. However, there are many variables included in this study.

We appreciate the reviewer comment regarding the paper being interesting.

-  Title is confusing because periodontitis affects alveolar bone, not only gingival tissues. So, please omit… a multifactorial and complex disease of oral gingival tissues.

We have changed the title to “Phenotypic and functional alterations of immune effectors in periodontitis; a multifactorial and complex oral disease” to encompass other tissues within the oral cavity which the reviewer so astutely has pointed out.

-Define PBMCs in the abstract.

We have defined PBMC in the abstract.

- Although “Periodontitis patients’ oral blood exhibited higher percentages of NK cells… in comparison to their peripheral blood”, it is important to highlight that there is not difference in the CD16+ NK percentage in oral blood between healthy and periodontal patients. Considering that there is a difference in the age of periodontal (29-68 years) and healthy patients (27-46 years), could the lower percentage of NK cells in peripheral blood be a matter of aging?.

We have added as suggested in the text.

We have analyzed the NK percentages within different age groups and did not see significant differences in the proportions of NK cells.

-There is not difference in the percentage of NK cells in Oral gingival-derived immune cells. Please argue this in the discussion.

We have discussed the point in the discussion section as suggested.

- “Since immune effectors in the periodontal tissues interact with a variety of stromal cells including epithelial cells”. In this way it seems that epithelial cells are stromal cells. Please, rephrase this sentence.

We changed the sentence as instructed.

-Considering that “The increased activation of orally derived mononuclear cells from patients could be due to their exposure to periodontal pathogens….”, this reviewer consider that one of the main points of this paper is the effect of periodontal environment on immune cells. Given that bacteria contribute to the development of periodontal disease, please evaluate the effect of F. nucleatum coincubation on orally derived immune cells (mononuclear cells) from healthy patients. Determine pro-inflammatory cytokines levels.

Since we did not observe differences between OBMCs and PBMCs from healthy individuals in terms of cytokine secretion, and we get very slight levels of pro-inflammatory cytokines in healthy OBMCs similar to PBMCs (Fig. 3) we did not carry out this assessment. Indeed, we do show the effect of F. nucleatum on PBMCs of healthy individual (Fig. 6 and S5).

Reviewer 3 Report

This manuscript describes that orally derived immune effectors having different survival capacity and functions compared with those derived from peripheral blood are possibly relevant to the pathogenesis of periodontal disease.  This study is relatively well designed and conducted.  However, this manuscript needs to be addressed to some points as described below.

General comments:

As the periodontal pathogenic bacterium, Fusobacterium nucleatum after the several treatments, such as fixation with paraformaldehyde and boiling and incubation with pronase, used in this study.  Please describe the reason why F. nucleatum was selected among a number of periodontal pathogenic bacteria, not from Porphyromonas gingivalis which is belonged in red complex, and live F. nucleatum was not used.

Please describe the reason why F. nuceleatum was treated with pronase.

Please describe the procedure of inactivation of pronase before the addition to cultured cells because pronase probably affects the viability and activities of cultured cells.

Please describe the reason about the increased cell death of peripheral blood mononuclear cells (PBMCs), especially periodontitis patients, compared to oral blood mononuclear cells (OMBCs) in more detail.

Please describe the mechanism and reason why NF-κB deletion increases IFN-γ production and decreases IL-6 production in more detail.

Regarding Figure 3A, please describe the reason why OMBCs from periodontitis patients did not increase the production of TNF-α by the stimulation with IFN-γ in more detail.

In Figure 3A, the TNF-α production level from untreated OMBCs from periodontitis patients is about 400 pg/ml, but that level shown in Figure 3B is 8,000 pg/ml, which is approximately 10-fold value than that in Figure 3A.  Please explain this difference.

Lines 333-337: There is no symbols showing significant differences of the secretion level of TNF-α in allogenic healthy PBMCs after the incubation with OMBCs-derived supernatants between from periodontitis patients and healthy individuals shown in Figure 4C.

Lines 339-342: There is no symbols showing significant differences of the secretion levels of TNF-α and IL-1β in allogenic healthy IL-2 treated PBMCs after the incubation with OMBCs-derived supernatants between from periodontitis patients and healthy individuals shown in Figures 4C and 4D.

Figure 3 shows that the secretion level of TNF-α in OMBCs-derived supernatants is approximately 400 – 8,000 pg/ml.  By the addition of OMBCs-derived supernatants to the culture of PBMCs, high amount of TNF-α is carried into the culture of PBMCs.  Please describe the reason why the concentration of TNF-α is below 100 pg/ml as shown in Figure 4C.

Lines 383-385: In Fig. 6A, there is no symbol showing a significant difference of the secretion levels of IFN-γ in co-cultured of PBMCs with between HEp2-IκB tumors and HEp2-pRcCMV control, in absence of F. nucleatum.

Lines 388-390: In Fig. 6B, there is no symbol showing a significant difference of the secretion levels of IFN-γ in co-cultured of IL-2-treated NK cells with between HEp2-IκB tumors and HEp2-pRcCMV control, in absence of F. nucleatum.

Lines 401 -403: In Figure S6, there is no symbol showing a significant difference of the secretion levels of GM-CSF and IL-13 in IL-2-treated NK cells co-cultured with HEp2-pRcCMV cells between absence and presence of F. nucleatum.

Figure S5: Please explain the reason why there are no significant differences of the concentration of MCP-1 and RANTES in PBMCs co-cultured with HEp2-IκB (S32AS36A) between in presence and absence of F. nucleatum.

Figure S5: Please explain the reason why there are no significant differences of the concentration of MCP-1 and IL-6 in PBMCs (-F. nucleatum) co-cultured with between HEp2-pRcCMV control and HEp2-IκB (S32AS36A).

Please explain the reason why the concentration of RANTES in NK cells were not determined in Figure S6 experiment.

Specific comments:

Line 70: “F. nucleatum” should be italicized.  Please check this issue throughout the manuscript.

Lines 141-143: Please describe the information about gender of blood donors.

Table 1 has no symbols showing the significant differences described in the text and its legend.

Table 2: Regarding the numbers of CD3+, CD19+ cells and CD3+CD4+ cells, are there no significant differences between the healthy individuals and periodontitis patients?

Line 301 in Figure 2 legend: “**(p value 0.001-0.01)” should be deleted because Figure 2 has no “**” symbol.

Lines 320 and 324 in Figure 3 legend: Please change “overnight” to “12 to 18 hours-“ as described in the text.

Line 386: Please change “GMCSF” to “GM-CSF” and add “GM-CSF: Granulocyte macrophage colony-stimulating factor” in the Abbreviations list.

Line 396: Is “NK cells” correct?  “IL-2-treated NK cells”?

Lines 410 and 412: Please change “left treated” to “left untreated”.  Please check this issue in Figure S5 and S6 legends.

Line 414: Please insert “at” between “cells” and “1:1 ratio”.  Please check this issue in Figure S6 legends.

Line 415: Please describe about “**” symbol.

Line 461: Please change “IL1β” to “IL-1β”.

“rhIL-2: Recombinant human IL-2” should be deleted because of no description in the text.

Figure 6 have no legend showing diagonal bars and black bars.

Please change “Figure 6” to “Figure S6” in the legend title.

Please change “Hep2” to “HEp2” in Figure S6 legend.

Figure S3: There is no numbers in each quadrant representing the percentages of stained cub-population for the specific antigen.

Figure S4A and S4B: Please change “TNF-a” on X-axis to “TNF-α”.

Figure S5 have no marks (box) showing PBMCs (-F. nucleatum) and PBMCs (+F. nucleatum).

Author Response

We are grateful to this reviewer because of his/her thorough and complete review. We apologize for the omissions and are thankful for his/her greatly constructive review which has helped us to improve the paper significantly.

This manuscript describes that orally derived immune effectors having different survival capacity and functions compared with those derived from peripheral blood are possibly relevant to the pathogenesis of periodontal disease.  This study is relatively well designed and conducted.  However, this manuscript needs to be addressed to some points as described below.

We appreciate the kind words of the reviewer about well-designed and conducted study.

General comments:

As the periodontal pathogenic bacterium, Fusobacterium nucleatum after the several treatments, such as fixation with paraformaldehyde and boiling and incubation with pronase, used in this study.  Please describe the reason why F. nucleatum was selected among a number of periodontal pathogenic bacteria, not from Porphyromonas gingivalis which is belonged in red complex, and live F. nucleatum was not used.

We apologize for the mistake, during cutting and pasting from our previous publications the wrong information was included in the materials and methods section. We have corrected the information in the revised manuscript. We have used both viable and paraformaldehyde fixed F. nucleatum in the treatments. Also since we have a great amount of knowledge regarding F. nucleatum and have published several papers on it we wanted to build on the previous studies in order to advance knowledge quickly

Please describe the reason why F. nuceleatum was treated with pronase.

We apologize for the mistake. Please see the answer above. We have removed the section on pronase.

Please describe the procedure of inactivation of pronase before the addition to cultured cells because pronase probably affects the viability and activities of cultured cells.

We apologize for the mistake. Please see the answer above. We have removed the section on pronase treatment.

Please describe the reason about the increased cell death of peripheral blood mononuclear cells (PBMCs), especially periodontitis patients, compared to oral blood mononuclear cells (OMBCs) in more detail.

We have included a paragraph in discussion as suggested.

Please describe the mechanism and reason why NF-B deletion increases IFN-g production and decreases IL-6 production in more detail.

We have included a paragraph in discussion as suggested.

Regarding Figure 3A, please describe the reason why OMBCs from periodontitis patients did not increase the production of TNF-by the stimulation with IFN- in more detail.

We have included a paragraph in result section as suggested

In Figure 3A, the TNF- production level from untreated OMBCs from periodontitis patients is about 400 pg/ml, but that level shown in Figure 3B is 8,000 pg/ml, which is approximately 10-fold value than that in Figure 3A.  Please explain this difference.

Since the amount of blood we obtain from oral blood is very small in comparison to the peripheral blood we could not run multiple experiments with the same oral blood. Therefore, the results in Fig. 3A and 3B is obtained from different donors, and we know that there is significant variability in the levels of secretion between the patient donors. However, the trends are always similar in terms of patients having higher induction levels when compared to healthy oral blood.

Lines 333-337: There is no symbols showing significant differences of the secretion level of TNF- in allogenic healthy PBMCs after the incubation with OMBCs-derived supernatants between from periodontitis patients and healthy individuals shown in Figure 4C.

We have now included the significance for the difference.

Lines 339-342: There is no symbols showing significant differences of the secretion levels of TNF- and IL-1 in allogenic healthy IL-2 treated PBMCs after the incubation with OMBCs-derived supernatants between from periodontitis patients and healthy individuals shown in Figures 4C and 4D.

As stated above we have now included the significance for the difference for TNF-a, but we do not see significance for IL-1b. We have avoided including lines where there is no significance in the results in order to avoid making the figures very dense.

Figure 3 shows that the secretion level of TNF- in OMBCs-derived supernatants is approximately 400 – 8,000 pg/ml.  By the addition of OMBCs-derived supernatants to the culture of PBMCs, high amount of TNF- is carried into the culture of PBMCs.  Please describe the reason why the concentration of TNF- is below 100 pg/ml as shown in Figure 4C.

As stated above since the amount of blood we obtain from oral blood is very small in comparison to the peripheral blood we could not run multiple experiments with the same oral blood. Therefore, the results in different figures are obtained from different donors, and we know that there is significant variability in the levels of secretion between the patient donors. However, the trends are always similar in terms of patients having higher induction levels when compared to healthy oral blood.

Lines 383-385: In Fig. 6A, there is no symbol showing a significant difference of the secretion levels of IFN- in co-cultured of PBMCs with between HEp2-IB tumors and HEp2-pRcCMV control, in absence of F. nucleatum.

We have now included the significances for the differences.

Lines 388-390: In Fig. 6B, there is no symbol showing a significant difference of the secretion levels of IFN- in co-cultured of IL-2-treated NK cells with between HEp2-IB tumors and HEp2-pRcCMV control, in absence of F. nucleatum.

We have now included the significances for the differences.

Lines 401 -403: In Figure S6, there is no symbol showing a significant difference of the secretion levels of GM-CSF and IL-13 in IL-2-treated NK cells co-cultured with HEp2-pRcCMV cells between absence and presence of F. nucleatum.

We have now included the significances for the differences.

Figure S5: Please explain the reason why there are no significant differences of the concentration of MCP-1 and RANTES in PBMCs co-cultured with HEp2-IB (S32AS36A) between in presence and absence of F. nucleatum.

Here we also observe plateauing effect at the highest concentation of induction in the presence and absence of F. nucleatum, thus the reason for no significant differences

Figure S5: Please explain the reason why there are no significant differences of the concentration of MCP-1 and IL-6 in PBMCs (-F. nucleatum) co-cultured with between HEp2-pRcCMV control and HEp2-IB (S32AS36A).

We do see significant differences and have included them in the figures.

Please explain the reason why the concentration of RANTES in NK cells were not determined in Figure S6 experiment.

We have a number of premixed cytokine arrays with different cytokine and chemokine mixed and when we run out of one panel we use an alternative panel which provides the most important assessments. We always prioritize on IFN-g and TNF-a since we have worked on these cytokines extensively and found that they are important in differentiation of the cells. We also focus on IL-6 since it is known to regulate IFN-g. The panel that we used for the assessment of NK cells had IL-8 as a representation of a chemokine and not MCP-1 and Rantes.

Specific comments:

Line 70: “F. nucleatum” should be italicized.  Please check this issue throughout the manuscript.

We thank the reviewer and have corrected all of the F. nucleatum as instructed throughout the manuscript.

Lines 141-143: Please describe the information about gender of blood donors.

We have included gender of the donors as suggested.

Table 1 has no symbols showing the significant differences described in the text and its legend.

We inserted symbols as instructed

Table 2: Regarding the numbers of CD3+, CD19+ cells and CD3+CD4+ cells, are there no significant differences between the healthy individuals and periodontitis patients?

Although we see a great difference in CD19+ cells in terms of mean percentage values but because of the variability among donors the significance was 0.14, however, no significant differences could be find for CD3+ and CD3+CD4+ cells.

Line 301 in Figure 2 legend: “**(p value 0.001-0.01)” should be deleted because Figure 2 has no “**” symbol.

We have deleted as instructed.

Lines 320 and 324 in Figure 3 legend: Please change “overnight” to “12 to 18 hours-“ as described in the text.

We changed as instructed.

Line 386: Please change “GMCSF” to “GM-CSF” and add “GM-CSF: Granulocyte macrophage colony-stimulating factor” in the Abbreviations list.

We changed as instructed.

Line 396: Is “NK cells” correct?  “IL-2-treated NK cells”?

We have changed to IL-2 treated NK cells.

Lines 410 and 412: Please change “left treated” to “left untreated”.  Please check this issue in Figure S5 and S6 legends.

We have changed as instructed.

Line 414: Please insert “at” between “cells” and “1:1 ratio”.  Please check this issue in Figure S6 legends.

We changed as instructed.

Line 415: Please describe about “**” symbol.

We inserted the symbols as instructed

Line 461: Please change “IL1” to “IL-1”.

We changed as instructed.

“rhIL-2: Recombinant human IL-2” should be deleted because of no description in the text.

We have deleted rhIL-2

Figure 6 have no legend showing diagonal bars and black bars.

We have inserted the type of bars in the legend of Figure 6.

Please change “Figure 6” to “Figure S6” in the legend title.

We have changed as instructed.

Please change “Hep2” to “HEp2” in Figure S6 legend.

We changed as instructed.

Figure S3: There is no numbers in each quadrant representing the percentages of stained cub-population for the specific antigen.

Because greater than 99% of the NK and CD8+ T cells are CD69 positive when they are activated with PMA/I we omitted the percentages to demonstrate the dots clearly in the respective quadrant. Please note when NK cells are activated with PMA/I they downmodulate the CD16 receptor expression.

Figure S4A and S4B: Please change “TNF-a” on X-axis to “TNF-”.

We changed as instructed.

Figure S5 have no marks (box) showing PBMCs (-F. nucleatum) and PBMCs (+F. nucleatum).

We have added the boxes in the legend for Fig. S5 and S6

Reviewer 4 Report

This paper attempts to find the potential similarities and differences between peripheral, oral and tissue derived immune effectors. Initially, the author obtained and characterized immune subsets from healthy individuals and from the patients with periodontal disease. The author discussed that the relative numbers of immune subsets obtained from peripheral blood may not represent the composition of the immune cells in the oral environment and that orally derived immune effectors may differ in survival and function from those of peripheral blood. Obviously, the orally derived immune effectors contain elements of saliva in addition to oral microorganisms, and thus exhibit phenotypic and functional properties which are distinct from those obtained from peripheral blood or even those found in the gingival tissues.

Results are well presented and fruitful to understand the immune effectors in healthy individuals and patients with periodontal disease. Specifically, NK cells are the regulators of adaptive immunity and the IFN-γ is primarily produced by activated T and NK cells which play an important role in host defense; and the use of NK cells, retroviral transduction, transfection and the generation of tumor cell transfectants and the Fusobacterium nucleatum preparation to support the findings of the study are promising.

The results are well discussed and the author mentioned that some of the possibilities that might be suspected from their results are under their investigation. For instance, the possibilities of decreased activation and release of IFN-γ during chronic inflammation by the NK cells are contentious in cytokine pathway, are under investigation. We are looking forward to see this result. Overall, the manuscript is written well, however, I have some minor concerns regarding this manuscript:

  • The last paragraph of introduction seems more descriptive rather than describing the background information of the study. It is recommended to describe the findings of the study in the result part (line 85-92).
  • It would be better if author explained about IL-2 and PMA/I in introduction like what their functions are.
  • In line 539-540, the author discuss about secreting IFN-γ, NK cells will be able to increase lysis of bacteria by augmenting the activation of monocyte-macrophages and dendritic cells. Is this authors’ finding? / Reference is required.
  • More detailed explanations are required for specification of classified patients.

Author Response

This paper attempts to find the potential similarities and differences between peripheral, oral and tissue derived immune effectors. Initially, the author obtained and characterized immune subsets from healthy individuals and from the patients with periodontal disease. The author discussed that the relative numbers of immune subsets obtained from peripheral blood may not represent the composition of the immune cells in the oral environment and that orally derived immune effectors may differ in survival and function from those of peripheral blood. Obviously, the orally derived immune effectors contain elements of saliva in addition to oral microorganisms, and thus exhibit phenotypic and functional properties which are distinct from those obtained from peripheral blood or even those found in the gingival tissues.

Results are well presented and fruitful to understand the immune effectors in healthy individuals and patients with periodontal disease. Specifically, NK cells are the regulators of adaptive immunity and the IFN-γ is primarily produced by activated T and NK cells which play an important role in host defense; and the use of NK cells, retroviral transduction, transfection and the generation of tumor cell transfectants and the Fusobacterium nucleatum preparation to support the findings of the study are promising.

The results are well discussed and the author mentioned that some of the possibilities that might be suspected from their results are under their investigation. For instance, the possibilities of decreased activation and release of IFN-γ during chronic inflammation by the NK cells are contentious in cytokine pathway, are under investigation. We are looking forward to see this result. Overall, the manuscript is written well, however, I have some minor concerns regarding this manuscript:

We appreciate the kind words of this reviewer and appreciate the time and effort spent in reviewing our paper

  • The last paragraph of introduction seems more descriptive rather than describing the background information of the study. It is recommended to describe the findings of the study in the result part (line 85-92).

We appreciate the suggestion and have removed the lines 85-92 from the introduction

  • It would be better if author explained about IL-2 and PMA/I in introduction like what their functions are.

We have indicated in the introduction the use of IL-2 and PMA/I in the activation of the cells

  • In line 539-540, the author discuss about secreting IFN-γ, NK cells will be able to increase lysis of bacteria by augmenting the activation of monocyte-macrophages and dendritic cells. Is this authors’ finding? / Reference is required.

We have inserted the reference.

  • More detailed explanations are required for specification of classified patients.

We have now included the sex and age of the donors.

Round 2

Reviewer 3 Report

The paper is significantly improved. 

Please carefully check the symbols showing significant differences among multiple groups in all Figures before publication.

Author Response

The paper is significantly improved. 

Response: Thank you very much. We appreciate your time and efforts.

Please carefully check the symbols showing significant differences among multiple groups in all Figures before publication.

Response: Thank you again and we checked the symbols. We will check again during the author proof process.